# Modeling of a Wind Power System Using the Genetic Algorithm Based on a Doubly Fed Induction Generator for the Supply of Power to the Electrical Grid

**Abdelkarim Guediri [1], Messaoud Hettiri [2] and Abdelhafid Guediri [1,*]**

[1]   VTRS Laboratory, Faculty of Technology, University of El Oued, El Oued 39000, Algeria
[2]   LEVRES Laboratory, Faculty of Technology, University of El Oued, El Oued 39000, Algeria
*   Correspondence: toufikguediri1@gmail.com or abdelhafid-guediri@univ-eloued.dz

**Abstract:** This paper is interested in studying a system consisting of a wind turbine operating at variable wind speeds, and a two-feed asynchronous machine (DFIG) connected to the grid by a stator and fed by a transducer at the side of the rotor. The conductors are separately controlled for active and reactive power flow between the stator (DFIG) and the grid. The proposed controllers generate reference voltages for the rotor to ensure that the active and reactive power reaches the required reference values, to ensure effective tracking of the optimum operating point and to obtain the maximum electrical power output. Dynamic analysis of the system is performed under variable wind speeds. This analysis is based on active and reactive energy control. The new work in this paper is to introduce theories of genetic algorithms into the control strategy used in the switching chain of wind turbines in order to improve performance and efficiency. Simulation results applied to genetic algorithms give greater efficiency, impressive results, and stability to wind turbine systems are compared to classic PI regulators. Then, artificial intelligent controls, such as genetic algorithms control, are applied. Results obtained in the Matlab/Simulink environment show the efficiency of this proposed unit.

**Keywords:** doubly fed induction generator; variable speed wind turbine; genetic algorithm; maximum power point tracking; fuzzy logic controller; proportional integral

## 1. Introduction

Wind energy is one of the most promising sources of renewable energy in the world, and this is mainly due to the reduction in environmental pollution resulting from classical power plants, as well as the dependence on fossil fuels with limited reserves [1,2]. Electric power generated from wind power plants is the fastest developing and most promising renewable energy source. The environmental degradation of air is one of the major problems that has prompted authorities around the world to take a set of measures to reduce the emission of pollutants [3,4]. To adapt to these new restrictions, environmentally friendly energy such as wind energy has been promoted, and many wind plants have been established in the world, being the only method for capable of inexpensive and mass production.

The optimization procedure is a technique of great importance for dealing with decision-making problems. It has grown significantly with the great development of computer systems technologies in terms of processing capacity and speed [5,6]. In fact, optimization seeks to improve performance by approaching one (or more) ideal point among many possible points or solutions based on criteria dictated by the specifications of the systems considered [7]. It is one of the most important branches of modern applied mathematics, and a lot of practical and theoretical research has been devoted to it. The theory includes the quantitative study of optimums and methods for finding them [8]. The solution to an optimization problem involves exploring the search space in order to maximize (or reduce) a particular function.

The relative complexities (in size or structure) of the search space and functionality to be optimized lead to the use of radically different accuracy methods [9]. Optimization methods can be categorized in different ways: deterministic and non-deterministic methods (also called stochastic or stochastic research methods); the choice of this or that method depends on the system to be studied and its complexity [10]. Deterministic methods are characterized by their simplicity and speed. They are used in the case when the system to be improved has a simple structure [11]. However, the main disadvantage of these methods is that this simplicity decreases as the number of variables to be optimized increases and the system becomes complex.

Under these conditions, the solution can converge towards local solutions [12]. While stochastic methods are more efficient and effective methods, they use stochastic processes based on stochastic exploration of the space of possible solutions [13]. Among the latter, we find the genetic algorithm, which represents a rather rich and interesting family of stochastic optimization algorithms. This was inspired by the concepts of evolution and natural selection [14]. Thanks to probabilistic research based on the mechanism of natural selection and genetics, genetic algorithms are highly effective and powerful in a general set of problems. The genetic algorithm maintains a set of encoded solutions, and guides this set towards the optimal solution [15].

In fact, to find an optimal solution to a problem in a complex space, it is necessary to find a compromise between two goals: exploring better solutions and powerful exploitation of the search space. Analytical studies have shown that genetic algorithms optimally manage this trade-off [16]. The aim and scope of the research is to improve the active and ineffective capacity associated with the electrical network by means of the genetic algorithm in increasing the number of iterations, and also introduce modifications of kp and Ki.

The obtained results were applied to a very large capacity of 1.5MW, and it was difficult to control and get obtain results compared to weak capacities. Nevertheless, we obtained very satisfactory results compared to the regulators proportional integral and fuzzy logic controller in terms of error, response time, and ripples, and therefore, the results show this towards the end of this study.

The main contributions of this paper are summarized as follows:

- Study of a system consisting of a variable wind speed wind turbine and an asynchronous dual-feed machine (DFIG) connected to the grid by a stator and fed by a transducer.
- The response of the system was verified by applying the proposed regulator in terms of its effectiveness towards the active and reactive power.
- Improvement in the results obtained through previous published works, in terms of response time, accuracy, low error, and stability.

## 2. Definition of Optimization

An optimization problem is defined as the search for the minimum or the maximum (of the optimum) of a given function We can also find optimization problems for which the variables of the function to be optimized are constrained to evolve within a certain part of the search space. In this case, we have a particular form of what we call a constrained optimization problem [17].

## 3. Objective Function and Fitness

We refer to objective function as the function which we wish to optimize. Fitness is the evaluation function of the individual. The fitness function is determined according to the problem posed (to be optimized). In the framework of from a simple function optimization, the fitness function is the objective function [18]. Fitness can be thought of as a measure of profit, utility, or quality. It is used to attribute to an individual a numerical value in relation to the interest it represents as a solution. Individuals in a population will be selected or eliminated based on their fitness. Only individuals with high fitness will be reproduced.

## 4. Maximization of Power without Speed Control

The control model is predicated on the notion that at a steady state, the wind speed is marginally different. Here, we obtain [19]:

$$C_{aer} = Cp\frac{1}{2}\rho S \frac{1}{\Omega_{\text{turbine\_estimated}}} v^3_{\text{estimated}} \tag{1}$$

With:

$$v_{\text{estimated}} = \frac{\Omega_{\text{turbine\_estimated}} \cdot R}{\lambda} \tag{2}$$

where w denotes the wind turbine's rotational speed. The link between the tip-speed ratio and the power coefficient Cp is seen in Figure 1. It should be observed that a value of is present to guarantee a maximum of Cp. Accordingly, it may be said that there is a turbine rotational speed value for a given wind velocity that permits obtaining the most mechanical power feasible from the wind, and this is precisely the turbine speed to be followed [20].

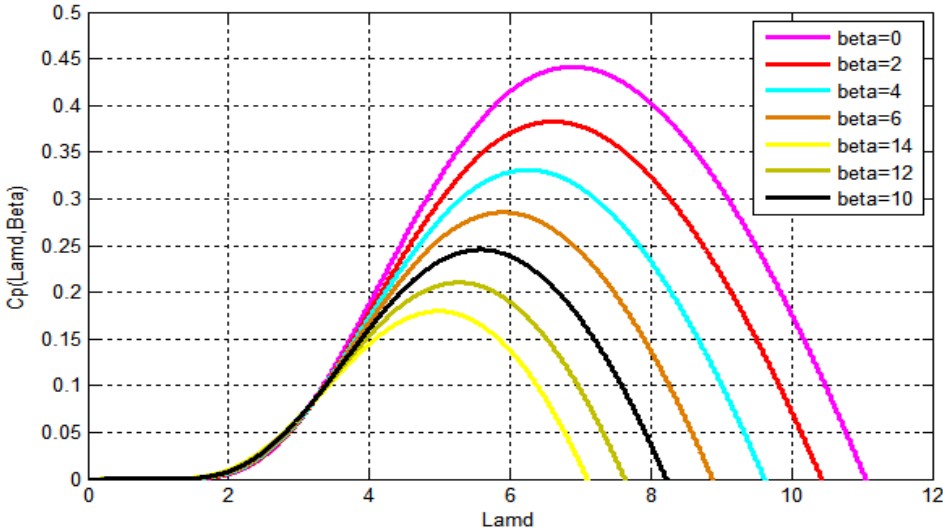

**Figure 1.** Typical Power Coefficient versus Tip-Speed-Ratio Curve.

The relationship between the punctuation speed and the angle of passage of the blades frequently determines the power coefficient. To support this thesis, we apply the formula given in the research papers for a 1.5 MW solar power plant as follows:

We set the speed ratio to the value ˇ$_{Cp\ max}$, which corresponds to the maximum power factor $C_{p\ max}$ and the reference torque, which is exactly proportional to the square of the generator speed, and may be obtained by combining the aforementioned formulae [21].

$$C_{em\_ref} = \frac{\rho\pi R^5}{2G^3} \frac{C_p}{\lambda^3 C_{pmax}} \Omega^2_{mec} \tag{3}$$

We disregard all of the losses in the converters and the filter, as well as the mechanical and Joule losses in the machine's stator and rotor. We can write in this situation [22]:

$$\begin{aligned}
\widehat{P}_{\text{aéro}} &= P_{\text{élec}} + \Delta p \\
\widehat{P}_{\text{aéro}} &= P_{\text{élec}} + p_{\text{frot}} + p_{Js} + p_{Jr}
\end{aligned} \tag{4}$$

$$p_{Js} = 3R_s I_s^2 = 3R_s\left(i_{ds}^2 + i_{qs}^2\right) \tag{5}$$

$$p_{Js} = 3R_r I_r^2 = 3R_r\left(i_{dr}^2 + i_{qr}^2\right) \tag{6}$$

$$P_{mec} = f\Omega^2 \tag{7}$$

## 5. Basic Structure of a Fuzzy Maximum Power Point Tracking Command

The perturbation and observation approach is only extended by the fuzzy logic command. The primary goal of this research is to apply the fuzzy Maximum Power Point Tracking (MPPT) command to maximize electrical energy extraction in the wind power conversion chain [23]. Figure 2 depicts the suggested fuzzy controller (CF) structure.

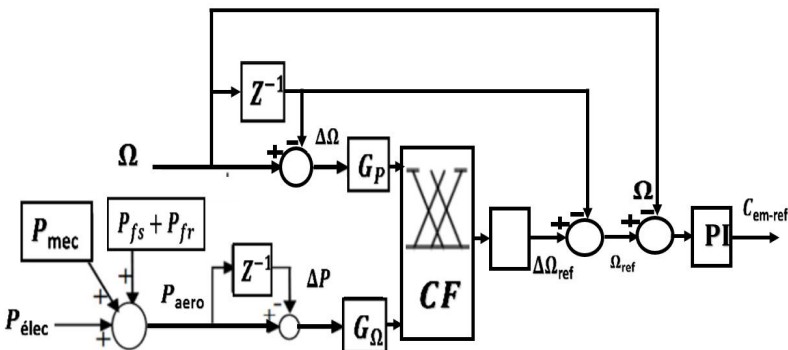

**Figure 2.** Structure of the fuzzy Maximum Power Point Tracking controller.

By neglecting losses of electrical origin, the electrical power becomes equal to the electromagnetic power defined by ($\Omega_{mec} \cdot C_{em}$). Because it is incompatible with the aerodynamic power and "respects the receiver convention of the assembly [24], this power (reference power) will not be positively counted. The wind turbine revolves at a fixed speed when these two powers are equal. The obtained results are shown in Figures 3–6.

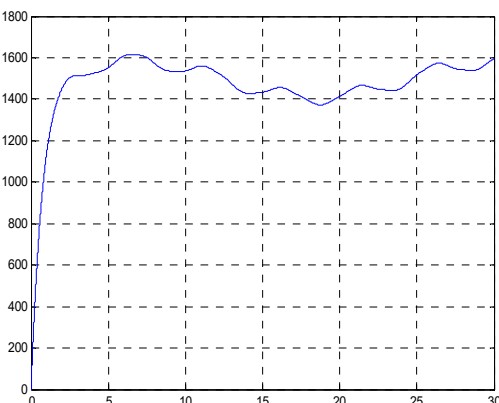

**Figure 3.** Mechanical speed (tr/min).

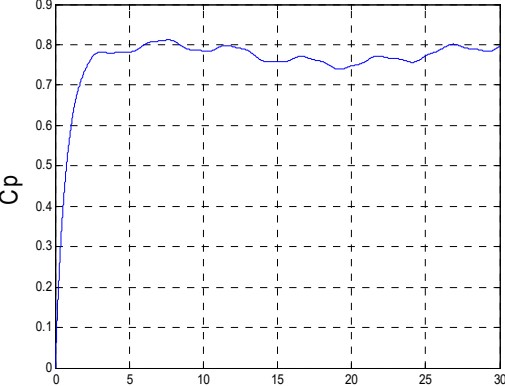

**Figure 4.** Power coefficient.

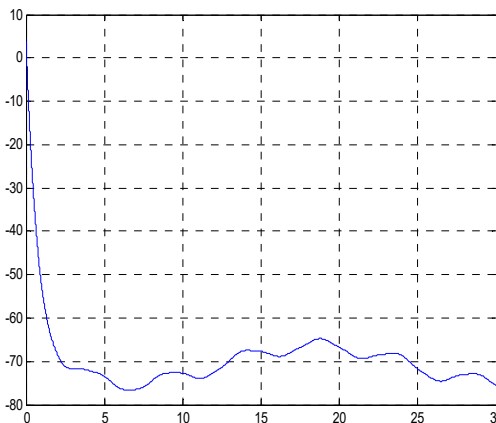

**Figure 5.** Torque produced.

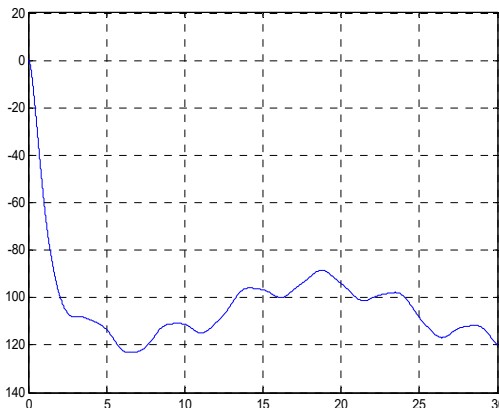

**Figure 6.** Electric power (N·m).

We see that the Cp reached an average value of 0.78; it aims to be maintained as much as possible in order to optimize output, and it varies somewhat depending on changes in wind speed.

## 6. Current Rotor Prediction Prediction

It is important to utilize both the direct and quadrature components of the rotor current in the following function [25]:

$$
\begin{aligned}
i_{dri}(k+1) &= \tfrac{T}{\sigma L_r}\left(V_{dri}(k) - r_r i_{dri}(k) + s_i \omega_{si} \sigma\, L_r i_{qri}(k)\right) + i_{dri}(k) \\
i_{qri}(k+1) &= \tfrac{T}{\sigma L_r}\left(V_{qri}(k) - r_r i_{qri}(k) - s_i \omega_{si} \sigma L_r i_{dri}(k) - s_i \tfrac{MV_s}{L_s}\right) + i_{qri}(k)
\end{aligned}
\tag{8}
$$

In addition, give the prediction of the active and reactive stator powers [26]:

$$
\begin{cases}
P_{si}(k+1) = -V_s \cdot \dfrac{M}{L_s} \cdot i_{qri}(k+1) \\
Q_{si}(k+1) = \dfrac{V_s \cdot \Phi_S}{L_s} - \dfrac{V_s \cdot L_m}{L_s} \cdot i_{dri}(k+1)
\end{cases}
\tag{9}
$$

## 7. System Description

The first double-fed induction generator setup employed in this study is shown in Figure 1, and it is physically coupled to the wind turbine using a gearbox and coupling shaft mechanism [26]. The stator and rotor of the wound-rotor induction generator are supplied separately from each component. The rotor is fed through the lower back to additional four-quadrant PWM power converters (RSC and GSC) linked with the usage of a battery within the direct current -link condenser, while the stator is directly connected to the grid [27].

A schematic illustration of a typical computer is shown in Figure 7 for converting commerce forces between a doubly fed induction generator, converters, and the grid. According to the reference torque provided by the Maximum Power Point Tracking control, the rotor aspect converter ensures a decoupled active and reactive stator power control, Ps and Qs Maximum Power Point Tracking. The grid facet converter controls how power flows via the grid. The rotor does this by keeping the direct current bus at a constant voltage level and by imposing a reactive energy QL of zero [28]. In Figure 8, the DC bus voltage is shown as a harmonic spectrum:

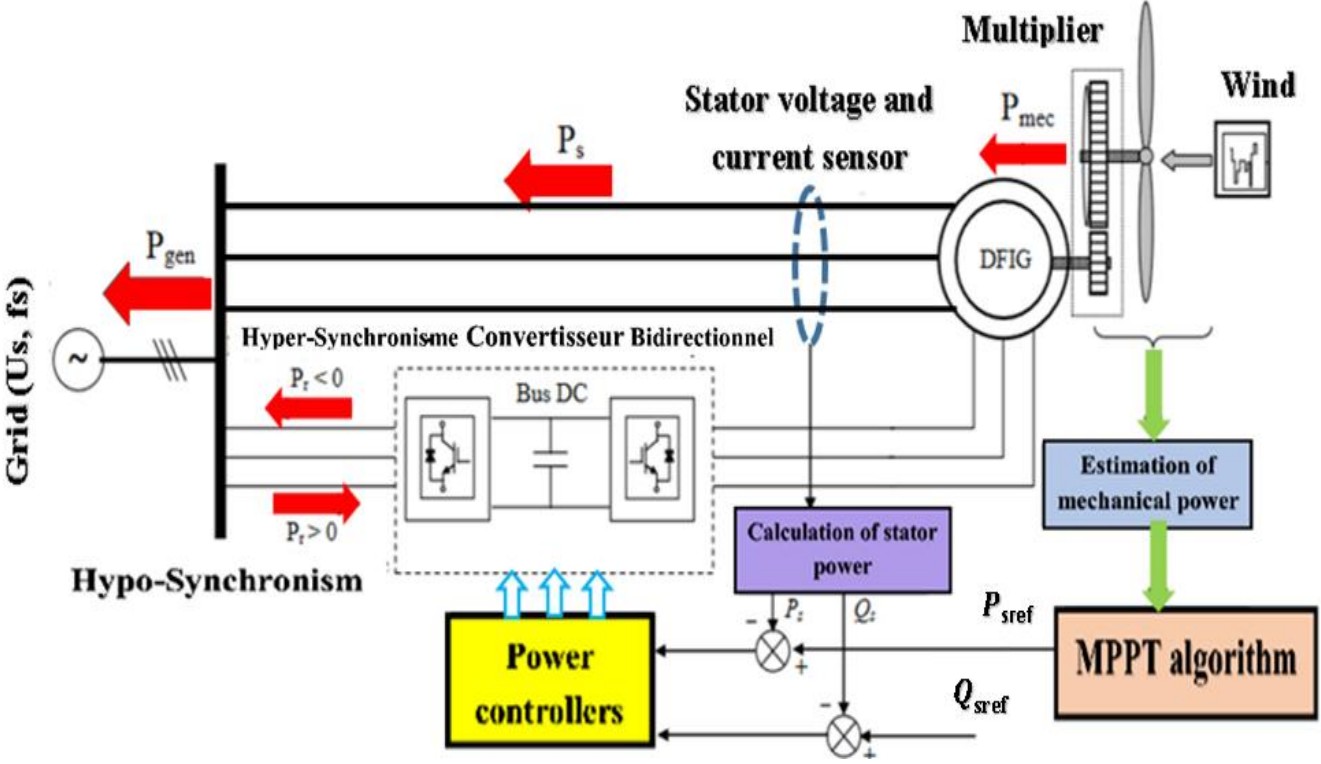

**Figure 7.** Configuration of a controlled wind power system connected to the grid.

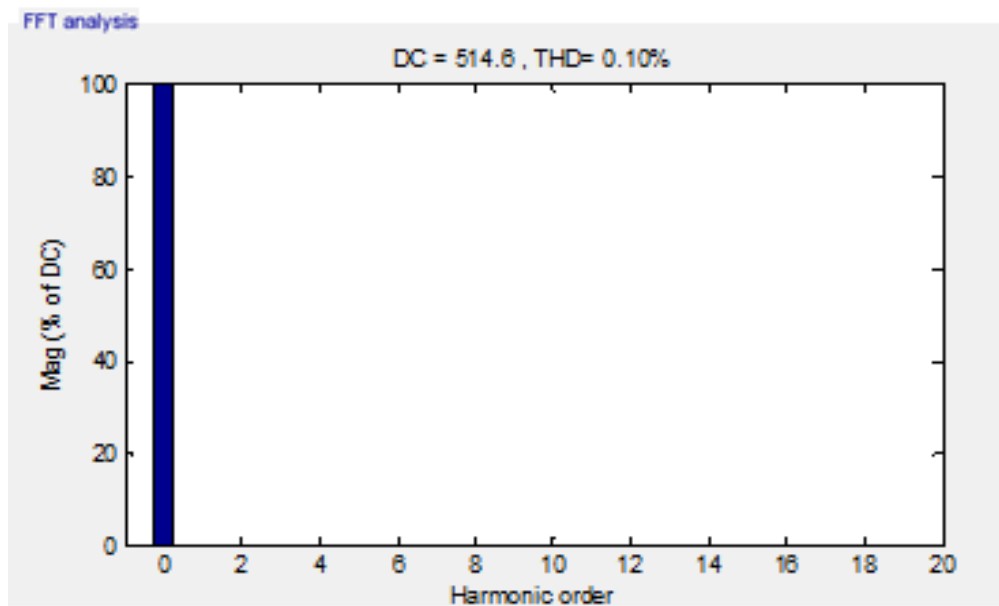

**Figure 8.** DC bus voltage control harmonics spectrum.

Additionally, the ripples of this voltage are quite tiny, and as seen in Figure 8, their harmonic spectrum exhibits a greatly decreased Total Harmonic Distortion (THD) of 0.10%. Additionally, this DC voltage maintains stability across the whole range of the wind profile fluctuation, ensuring a constant power flow between the grid and the generator's rotor.

Figure 9 represents the evolution of the continuous vector voltage, which shows the following:

❖ The DC bus voltage responds faster and without overshoot, reaching the set value of 514.6 V.

❖ The shape of the DC vector voltage is smoother, which has the advantage of changing the wind speed.

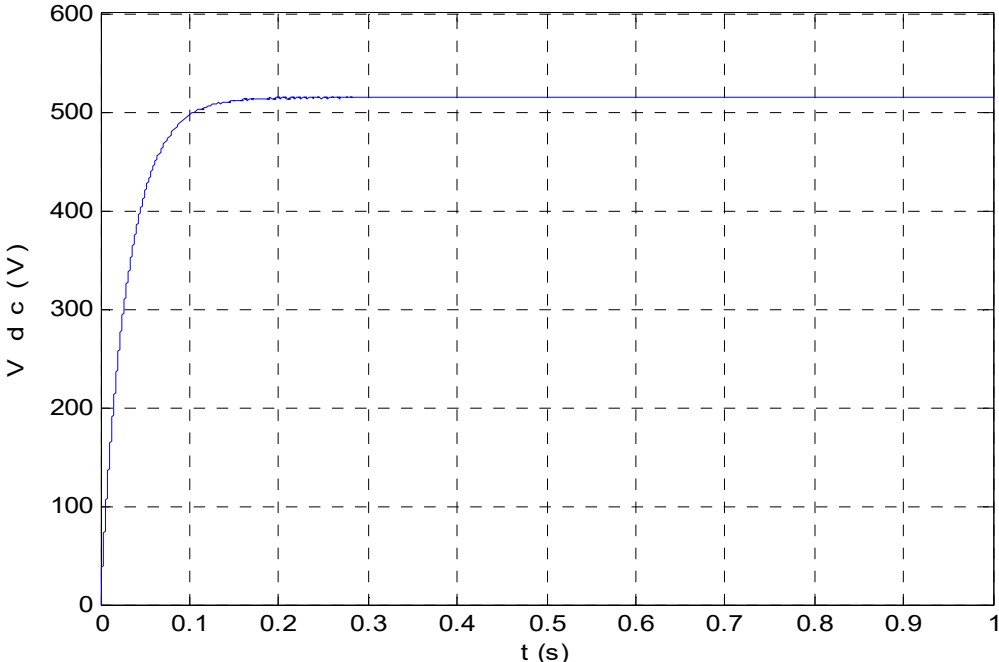

**Figure 9.** DC bus voltage (V).

## 8. Arithmetic Crossing (Barycentric)

The inventor of this method is "MICHALEWICZ". We select the exchange places for this kind of crossover at random, followed by an arithmetic mean weighted by a coefficient a. Two children (offspring) $E_1(i)$ and $E_2(i)$ are produced when this procedure is carried out to the two parents $C_1(i)$ and $C_2(i)$, as shown in [29]:

$$\begin{cases} E_1(i) = a\,C_1(i) + (1-a)C_2(i) \\ E_2(i) = (1-a)C_1(i) + aC_2(i) \end{cases} \quad (10)$$

A non-uniform arithmetic crossing occurs when the value of an is randomly produced in the range $[-0.5; 1.5]$, as opposed to a uniform arithmetic crossing, where the value of an is a constant selected by the user. An illustration of the use of this sort of crossing is shown in the following picture [30]. Figure 10 shows the computational crossing method of the genetic algorithm.

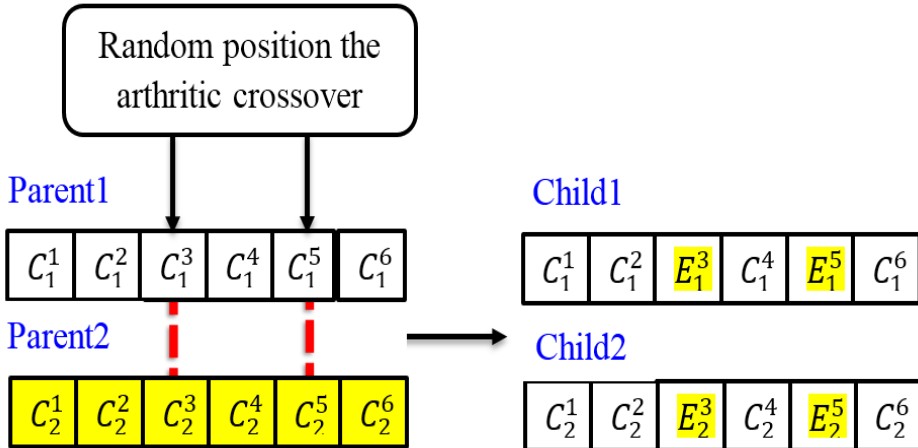

**Figure 10.** The arithmetic crossover.

This number indicates that the two new third genes appeared by [31]:

$$\begin{cases} E_1^3(i) = a\,C_1^3 + (1-a)C_2^3 \\ E_2^3(i) = (1-a)C_1^3 + aC_2^3 \end{cases} \tag{11}$$

## 9. Optimization of Doubly Fed Induction Generator Regulators by Genetic Algorithm

The use of conventional and fuzzy regulators to control the doubly fed induction generator stream yielded particularly satisfactory performance in dynamic mode. However, the main drawback noted is the lack of design techniques. In order to address this problem and improve the obtained performance, genetic algorithms were applied to design and optimize the gains of two conventional PI regulators and a fuzzy PI [32]. Genetic algorithms ensure this optimization and locate the global optimum. However, the combination of the genetic algorithm and a local search algorithm, such as Gradient or Simplex, allows the accurate calculation of the global optimum and improves the quality of the obtained result [33]. Two local research methods will be adopted as hybridization using genetic algorithms. The following two paragraphs briefly introduce the principle of each of these methods.

The optimization procedure is a hybrid algorithm which consists of a genetic algorithm combined with a local search method (Gradient or Simplex), and which acts on the parameters of the regulator [34]. The following figure shows the diagram of this procedure.

For every gene that mutates, we take numbers $\tau$ and. The first can take the values +1 for a effective alternate and −1 for a negative trade. The second is a randomly generated range within the variety [0 1]. It determines the value of the trade. Under those conditions, the $Ci'$ gene, which replaces the mutated gene, is calculated from one of the following relationships [35]:

$$\begin{cases} C_i' = C_i + (C_{max} - C_i)\left(1 - r^{\left(1 - \frac{G_F}{G_T}\right)^5}\right) & \text{if } \tau = +1 \\[4mm] C_i' = C_i - (C_i - C_{min})\left(1 - r^{\left(1 - \frac{G_F}{G_T}\right)^5}\right) & \text{if } \tau = -1 \end{cases} \tag{12}$$

where $C_{max}$, $C_{min}$ respectively denote the lower and higher limits of the price of the parameter $Ci$, and $GF \leq GT$ represents the era for which the amplitude of the mutation cancels out. Figure 11 illustrates and shows that.

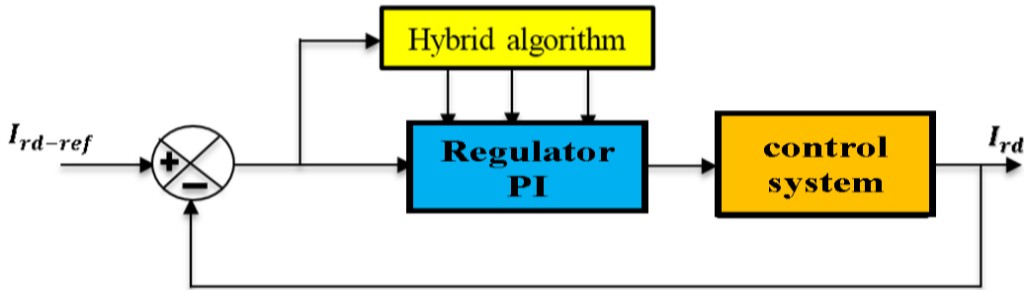

**Figure 11.** Principle of optimization using a genetic gradient or simplex algorithm.

Steps to implement the genetic algorithm on the PI regulator:

- In the first step, we choose the matrix at random, which contains a set of solutions.
- In the second step, through the value of KP and KI calculated in the PI regulator, we choose the KP domain and KI domain from the random matrix.
- In the third step, through the specified range, we choose the optimal and exact KP and KI values.

Using the following phases, the process for improving the regulator settings is described [36]:

- An initial offspring is randomly born.
- Evaluate this offspring.
- Apply genetic operators (selection, crossing, mutation).
- Evaluate the sort of the new offspring created through genetic operators.
- Repeat the process for a given variety of offspring.
- Choose the best character from the new offspring.
- Use a nearby seek approach (gradient or simplex) to finalize the optimization operation achieved by using the genetic algorithm.

## 10. Optimization of the Classic PI Regulator

With the help of a hybrid genetic set of simplex rules and the approach of the "Gatool" window that Matlab has invented, this regulator's optimization is carried out. The algorithm's inputs are listed below [37]:

- Size of the offspring T = 20.
- Selection using roulette.
- Multiple crossing with a chance *pc* = 0.8.
- Uniform mutation with opportunity *pm* = 0.01.
- Number of offspring N = 49.
- Hybridization technique: simplex.

## 11. Setting Genetic Algorithm Parameters:

The development of a genetic algorithm requires the adjustment of certain parameters. This setting has an influence on the convergence of the genetic algorithm and the results obtained. However, there is no specific rule for adjusting the parameters of a genetic algorithm (GA), and they are often empirically chosen. A few remarks should therefore be made [38,39].

- Probability of crossing: The probability of crossing has a considerable influence on the convergence speed of a genetic algorithm The greater it is, the more it promotes the recombination of individuals while promoting falling into an optimum local. Typical values for this parameter range from 0.6 to 0.95.
- Probability of mutation: It must be quite low compared to that of the crossing so as not to disturb the evolution of the algorithm A high value will transform the algorithm into a random search, while a very low value will make the extraction of local optima impossible. Typical values for this parameter range from 0.001 to 0.2.

*Mutation*

Mutation is a secondary operator; it prevents the premature stopping of the algorithm in a local solution. This operator is defined by a random bit value change in a chosen string with a low probability. The mutation adds a random search character to the genetic algorithm. The canonical form of the genetic algorithm is presented as follows [40,41]:

1. First step: Initialize all strings and genetic operators.
2. Second step: Evaluate fitness for each string.
3. Third step: Select pairs of strings of highest fitness value or using selection operator.
4. Fourth step: Create offspring using crossover and mutation.
5. Fifth step: Calculate fitness for offspring and check whether optimized solution is reached. If yes, exit, or otherwise go to step 3, as shown in Figure 12.

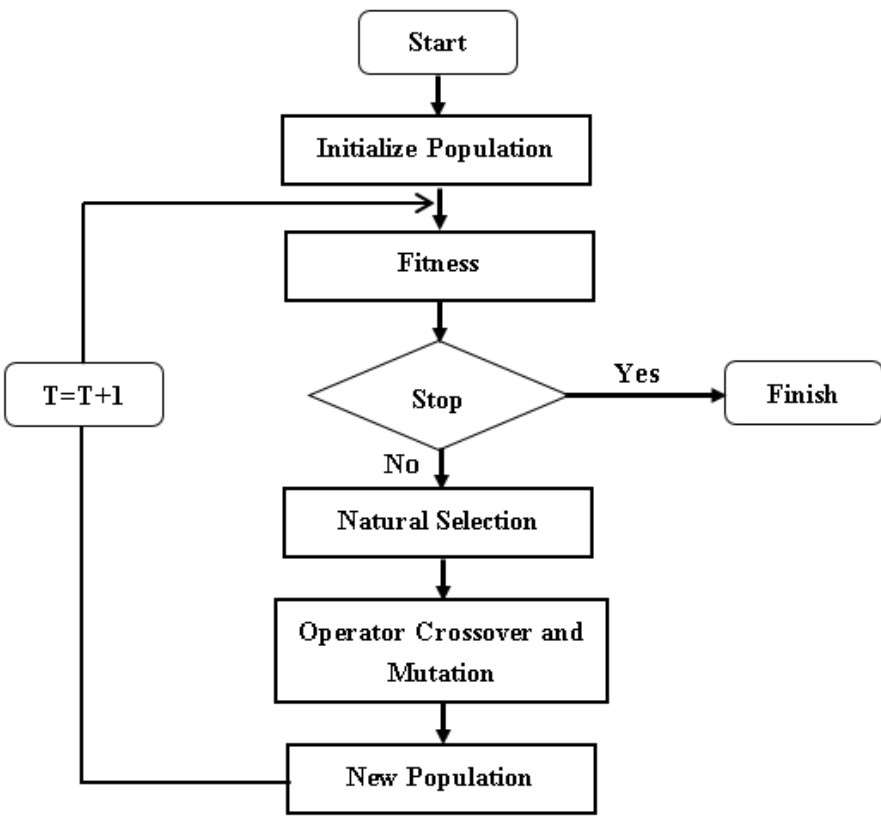

**Figure 12.** Schematic representation of the genetic algorithm.

## 12. Simulation Results and Discussion

The investigation was carried out on a 1.5 MW DFIG system the proposed genetic algorithm controls and parameters from DFIG and referred to in the Appendix A as Tables A1 and A2. Abbreviations, Greek symbols, and labels are also shown in Table A3. We carried out the identical simulation processes provided in this research in order to demonstrate the value of optimizing traditional PI gains using a genetic algorithm combined with the simplex approach. We saw an improvement in dynamic overall performance from the simulation results. According to the simulation described in "Genetic algorithm managing the wind electric machine in Figure 13", where we saw a notable improvement on the dynamic level compared to the PI regulators, the following outcomes were mostly dependent on the doubly fed induction generator:

For the robustness tests of the control by the genetic algorithm regulator, we studied the influence of the variation in the rotor resistance, own inductance, and mutual on the performance of the control. The simulation results of our wind power system (Turbine + DFIG) controlled by the genetic algorithm regulator.

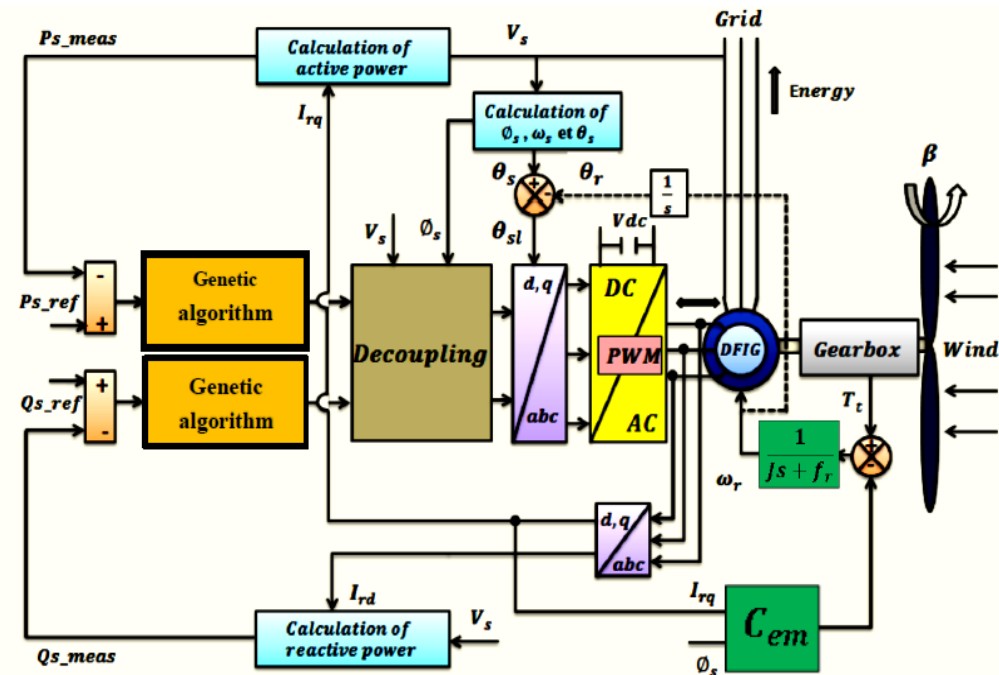

**Figure 13.** Global block diagram of the command of the genetic algorithm based on the doubly fed induction generator.

The starting is no-load, then a reference active power is applied:

- (Pref = 0 W); so that t ∈ [0 ; 0.2] s.
- (Pref = −20,000 W) Negative scale; so that t ∈ [0.2 ; 0.6] s.
- (Pref = −10,000 W); so that t ∈ [0.6 ; 1] s.
- Reactive power:
- (Qref = 0 VAR); so that t ∈ [0 ; 0.2] s.
- (Qref = −5000 VAR) Negative scale; so that t ∈ [0 ; 0.6] s.
- (Qref = 0 VAR); so that t ∈ [0.6 ; 1] s.

The figures below show the performance of the reactive and active stator power PI-genetic algorithm control applied to the doubly fed induction generator.

Figures 14 and 15 illustrate the responses of the system with the genetic algorithm controller. In general, it can be seen that the power steps are followed by the generator for both active and reactive power. However, we observe that the effect of the coupling appears on one of the two powers when changing the setpoint of the other power.

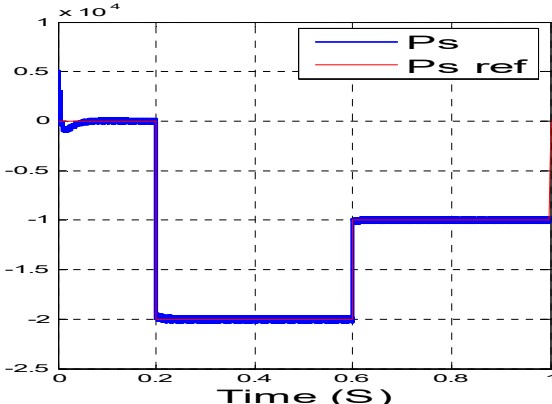

**Figure 14.** Active power stator.

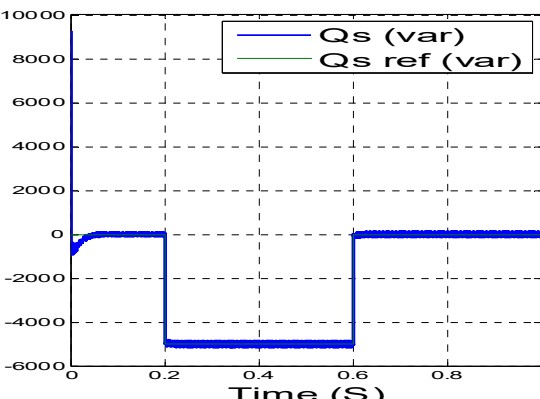

**Figure 15.** Reactive power stator.

We can determine the performance of these regulator in both transient and steady state using the following criteria:

- Maximum error (overshoot).
- The recovery or stabilization time (the response time).
- The residual error (the static error).

The forward and quadratic components of the rotor current are shown in Figure 16, and illustrate the control error of ird and irq. From these curves, we see that:

- The PI regulator maintains rotor currents at their respective references imposed by stator voltage regulation;
- A reduction in the load induces a reduction in the rotor current;
- The error in checking ird and irq is practically zero. The results obtained are illustrated in Figure 17. They show that the electromagnetic couple perfectly follows its benchmark with good dynamic performance, with less oscillation and overshoot.

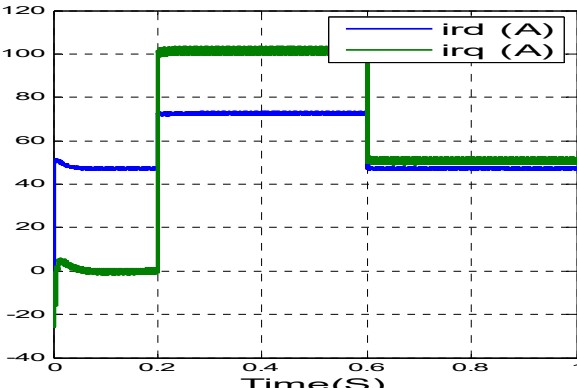

**Figure 16.** Direct currents and rotor quadrature.

Moreover, the results in Figures 18 and 19 illustrate the simulation results of the stator currents along the d and q axis and the three-phase stator currents generated by the doubly fed induction generator are proportional to the active power supplied. The waveform of the current is almost sinusoidal for both stator currents, which means good quality of power supplied to the grid. Figures 20 and 21 illustrate the simulation results of the stator current voltages at the terminals of the doubly fed induction generator and the control voltages of the rotor; the latter were obtained by a voltage inverter controlled by the genetic algorithm and which used the MLI technique. They show the waveform of the stator voltage and current. We can see that the stator voltage is equal to that of the grid, while the waveform of the current is related to that of the active power and the reactive power. The genetic algorithm regulator does not generate any overshoot, particularly at transient. For the other

performances, they are almost similar to that of the PI regulator, which shows regulation by genetic algorithm control excellence through the effective rejection of the effects of the disturbances from which the authorities completely trace their references.

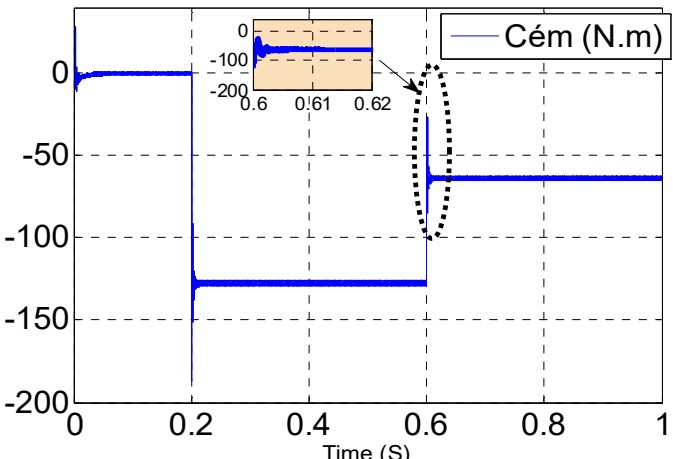

**Figure 17.** Electromagnetic torque.

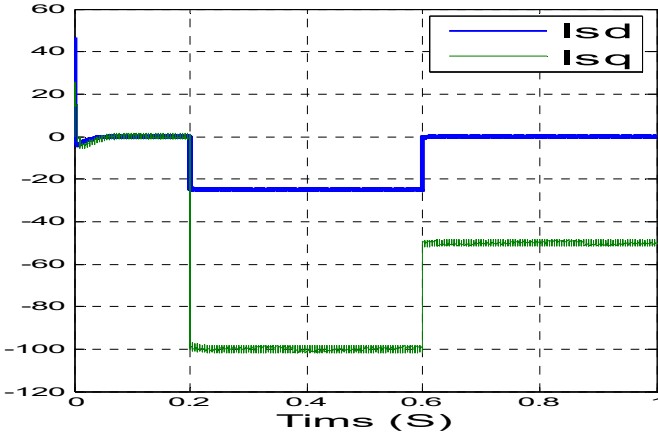

**Figure 18.** Direct currents and stator quadrature.

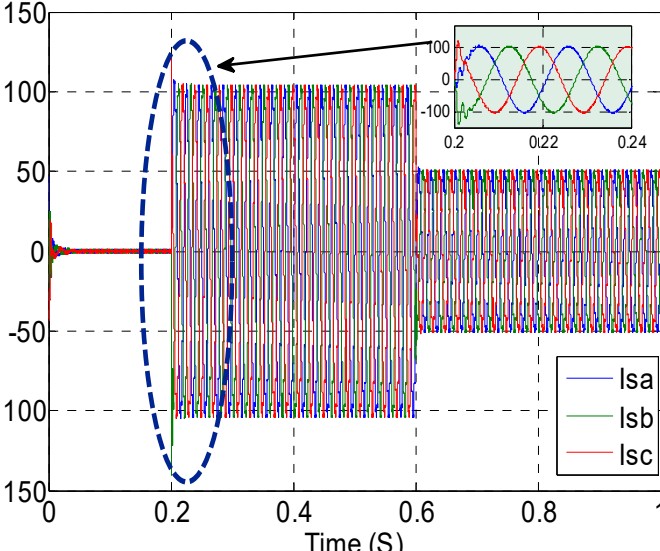

**Figure 19.** Stator three-phase currents (A).

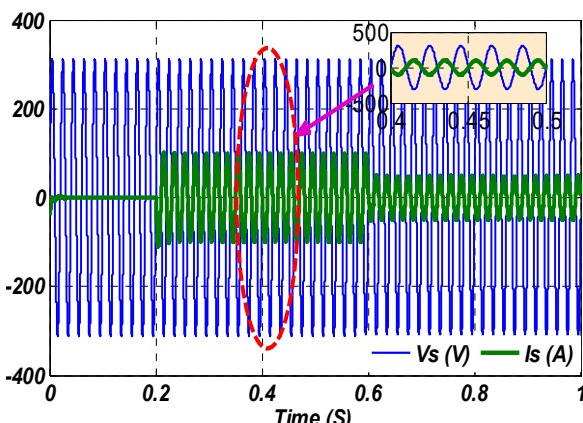

**Figure 20.** The stator current and voltage.

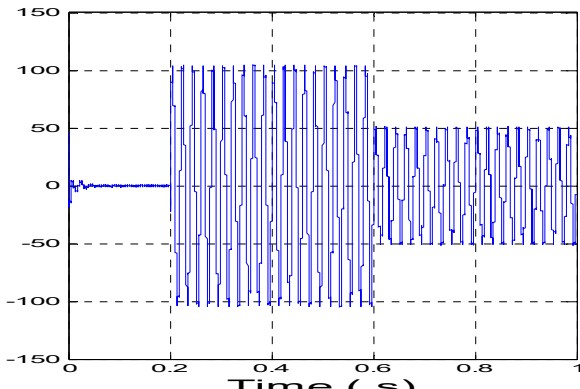

**Figure 21.** The stator current (A).

Figures 22 and 23 show an improvement in active and reactive energy in terms of performance using the combination genetic algorithm controller and compared with that of the conventional regulator; the evaluation criteria should be eliminated. These criteria must take into account both the maximum amplitude of the regulator error and the time required for the system to return to the set point after a perturbation, or to reach a new reference. The tuning using the genetic algorithm may override the tuning by the PI and Fuzzy logic controller regarding the dynamic response quality of the system. In effect, the latter reduces the response time by producing a finite overshoot accompanied by weak oscillations around the set point in a steady state, the accuracy is not as good as that of the regulator (PI and Fuzzy logic controller) where the integrated action cancels the static error. This then indicates a combination of the two types of regulators.

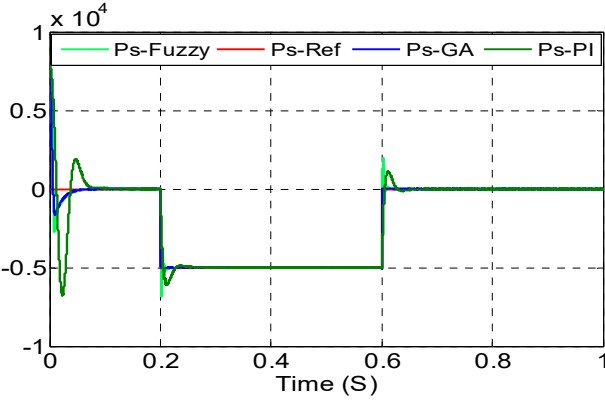

**Figure 22.** Active power stator (W).

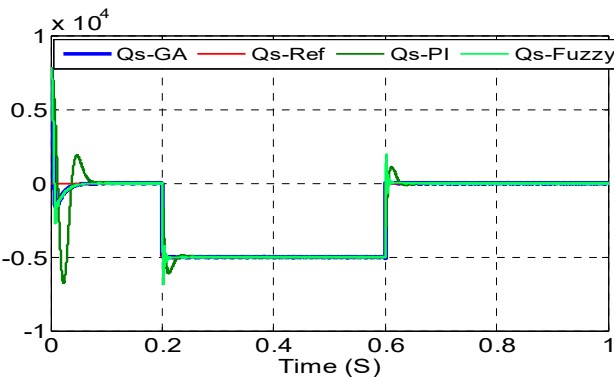

**Figure 23.** Reactive power stator (VAR).

- A genetic algorithm regulator: for the transient regime.
- A regulator (PI and Fuzzy logic controller): for the steady state.

The major disadvantage of genetic algorithm regulators is the matching of gains, ensuring system stability. In addition, the order is only calculated from two values: the error and the variation of the error. The genetic algorithm applied in this article has been proven to be very effective compared to the results published in the Indonesia Journal of Electrical Engineering and Computer Science under the title Optimization of PI Controller and Fuzzy logic controller Using Genetic Algorithm for Wind Turbine Application, as well as in the International Journal of System Assurance Engineering and Management under the title of fuzzy modeling and control of a wind power system based on a dual-feed asynchronous machine to supply power to the electric grid. The following points are:

- Response time.
- Precision.
- The error.
- Quality.
- Stability.
- Exceeding.
- Total Harmonic Distortion (THD).
- Sinusoidal.

## 13. Conclusions

In this paper, we proposed a genetic algorithm and an active and reactive gene algorithm connected to a stator network doubly fed induction generator (DFIG). The genetic algorithm efficacy was tested under different operating conditions, demonstrating optimization and efficiency in terms of duns against changing rotor resistance, insensitivity to torque disturbance, re-reducing response time, accuracy, speed or overtaking, large overrun reduction at start-up, and avoiding peak activity power, reduced power ripples, and improved Total Harmonic Distortion (THD), as well as faster dynamics with few stability errors in all dynamic operating conditions. The simulation results showed good control behavior oriented towards better performance of the proposed controller.

We can conclude that these are simple algorithms by design and can solve very complex problems with good accuracy. However, they have some limitations and difficulties. These difficulties depend on the choice of the stopping criteria: population size, number of generations, outbreeding and mutagenesis potential, and the techniques used to achieve them. The correct selection of these criteria requires a good knowledge of the system to be studied and the problem to be solved. Accordingly, it is possible to introduce other modern methods in addition to this algorithm to obtain better, more accurate and more efficient results for the system.

**Author Contributions:** The idea was proposed by A.G. (Abdelkarim Guediri) and M.H. wrote the paper, A.G. (Abdelkarim Guediri), M.H. and A.G. (Abdelhafid Guediri) contributed in the analysis and discussion of the results. All authors have read and agreed to the published version of the manuscript.

**Funding:** This research received no external funding.

**Institutional Review Board Statement:** Not applicable.

**Informed Consent Statement:** Not applicable.

**Data Availability Statement:** Not applicable.

**Conflicts of Interest:** The authors declare no conflict of interest.

## Appendix A

**Table A1.** Parameters of 1.5 MW doubly fed induction generator.

| Symbol | Parameters | Value |
|---|---|---|
| Pn | Rated Power | 1.5 MW |
| Vs. | Stator Voltage | 300 V |
| Fs | Stator Frequency | 50 Hz |
| Rs | Stator Resistance | 0.012 $\Omega$ |
| Ls | Stator Leackage Inductance | 0.0205H |
| Rr | Rotor Resistance | 0.021$\Omega$ |
| Lr | Rotor Leakage Inductance | 0.0204H |
| M | Mutual Inductance | 0.0169H |
| P | Pairs of poles number | 2 |
| J | Rotor inertia | 1000 Kg·m$^2$ |

**Table A2.** Parameters of Turbine.

| Symbol | Parameters | Value |
|---|---|---|
| R | Blade radius | 35.25m |
| N | Number of blades | 3 |
| G | Gearbox ratio | 90 |
| J | Moment of inertia | 1000 Kg·m$^2$ |
| $f_v$ | Viscous friction coefficient | 0.0024 N·m·s$^{-1}$ |
| V | Nominal wind speed | 16 m/s |
| $V_d$ | Cut-in wind speed | 4 m/s |
| $V_m$ | Cut-out wind speed | 25 m/s |

**Table A3.** Abbreviation.

| Nomenclature | | | |
|---|---|---|---|
| $\theta_{Sl}$ | Angle between the phase axis of the first stator winding and the rotor axis (rad) | $C_{em}$ | Electromagnetic torque (N·m) |
| $\theta_S$ | Angle between the axis of the first phase of the stator winding and the d axis (rad) | RSC | Rotor Side Converter |
| $\theta_r$ | Angle between the axis of the first phase of the rotor and the d axis (rad) | $Q_{S\_ref}$ | The reactive power at the reference stator (VAR) |
| $\omega_s$ | Electric stator pulse (rad/s) | $P_{S\_ref}$ | The active power at the reference stator (W) |
| G | Multiplier gain | $Q_{S\_meas}$ | The reactive power at the measured stator (VAR) |
| $f_r$ | Rotor feed frequency (Hz) | $P_{S\_meas}$ | The active power at the measured stator (W) |
| PWM | Acronyme Pulse with modulation | $C_{em\_ref}$ | Electromagnetic torque reference (N·m) |
| B(BITA) | Turbine blade pitch angle (rad) | $\lambda$ | Relative wind speed (m/s) |

**Table A3.** *Cont.*

| Nomenclature | | | |
|---|---|---|---|
| g | Slip | S | Area swept by the wind turbine rotor (m$^2$) |
| $\Omega_{mec}$ | Mechanical speed (rad/s) | $P_L$ | Active line power (W) |
| R | Turbine radius (m) | $Q_L$ | Line reactive power (VAR) |
| $V_{dc}$ | DC bus voltage (V) | $T_t$ | The turbine torque (N·m) |
| $\lambda_{opt}$ | Optimal speed ratio (m/s) | **Greek Symbols** | |
| $\omega_r$ | Electric rotor pulsation (rad/s) | $\rho$ | Air density at 15 °C (kg/m$^3$) |
| $I_{rd}$ , $I_{rq}$ | Rotor current along the d axis, q (A) | **Abbreviations** | |
| $\varphi_s$ | stator flux (Wb) | MPPT | Maximum Power Point Tracking |
| $P_s$ | Active stator power (W) | DFIG | Doubly-fed induction generator |
| $Q_s$ | Reactive stator power (VAR) | LSC | Line side converter |
| $\omega_t$ | Turbine speed (rad/s) | PI | Proportional Integral |
| $P_r$ | Active rotor power (W) | ref | Index indicating the reference (the setpoint) |
| $Q_r$ | Reactive rotor power (VAR) | DC/AC | Direct Current/Alternative Current |
| $V_s$ | Stator voltage vector (V) | THD | Total Harmonic Distortion |
| POP | Population | GA | Genetic algorithm |
| Mu | Mutation | CCR | Convertisseur Coté Rotor |
| FC | Fuzzy Controller | | |
| FGA | Fuzzy Génétique Algorithms | | |
| CCS | Convertisseur Coté Stator | | |
| GSC | Grid Side Converter | | |

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
