# Peer review of "Modeling of a Wind Power System Using the Genetic Algorithm Based on a Doubly Fed Induction Generator for the Supply of Power to the Electrical Grid"

_processes, doi:10.3390/pr11030952_

Round 1

Reviewer 1 Report

General statements about the article:
The manuscript "Modeling of a Wind Power System Using the Genetic Algorithm Based on DFIG for the Supply of Power to the Electrical Grid" is interesting and falls within the journal's subject. Furthermore, the majority of the references cited in the manuscript were published within the last four years. In my opinion, the manuscript can be of interest to the audience of the Processes; However, I indicated the following elements for revision:
1- I recommend adding the graphical abstract.
2- The abstract is entirely qualitative. Please add some important numerical data obtained from this study to the abstract.
3- For readers to quickly catch the contribution in this work, it would be better to highlight significant difficulties and challenges and your original achievements to overcome them in a clearer way in the abstract and introduction.
4- Some keyword is too long.
5- Avoid using abbreviations and acronyms in the title.
6- Some paragraphs are too long or too short. This negatively impacts the structure of the manuscript.
7- A table should be added to the end of the section "Introduction" to show the novelty and originality of the article.
8- Please check that the Figures taken from other articles have permission.
9- The obtained results have not been sufficiently compared with the published data. Please ‎add a Table in the "Results and discussion" section to address this issue.
10- The results should be validated with other similar studies.
11- The texts in figures 8, 11, and 12 are unclear.
12- The limitations of the study should be included and discussed.
13- Future directions should be added to the conclusion.
14- In the conclusion, I would appreciate a conclusion rather than a discussion. I would be satisfied with the conclusion, such as better than..., faster than..., lower computation effort.... with parameters.
15- Please avoid using words like "we," "us," "our," etc.

Author Response

First of all, the authors wish to thank the editor and the reviewers for their efforts in reviewing the paper and for their valuable comments and recommendation given to improve the quality of the paper. The authors hope that the revised paper is improved to their satisfaction. Response to the specific comments and suggestions is given below. Revisions made in the manuscript are highlighted in the yellow color.

Reviewer 2 Report

You need to carefully revise your manuscript in order to be able for publication on processes. 

Some of my major comments:

1. There is no actual state-of-the-art section.

2. There is no innovation mentioned.

3. There is no clear aim and scope of the research.

4. The methods are too complicated for a non-relevant reader.

5. The results section need to be rewritten in a more formal and concise manner.

6. The conclusions' section is inadequate and need to be reshaped and more information should be provided.

7. There are no future recommendations.

Fix these issues and then I would be able to recommend this paper for publication. 

Author Response

(The authors gave the same response as above.)

Reviewer 3 Report

This paper investigates how GA is applied in improving wind power output performance. It is an interesting and important topic. However, the organization and writing of this paper need careful improvement, to reach the publication standard as academic research. 

1. It is not clear what information needs to be revealed in fig.1? You need to show clearly what is the purpose of such theoretical analysis, to support your following content? Similar unclear introduction for Figure 8.

2. Line 151: why it is believed a smooth vector voltage is linked to the advantage of changing wind speed? In other words, how to guarantee DC output at another wind speed is still stable? or even if it is not stable, you can still adjust the rectifier when changing wind speed?

3. Sections 6-7: you simply introduced the GA principle, what is the difference or innovative improvement you made during calculation?

4. section 8- section 9: It is not clear what each factor means in your own simulation, you need to indicate in your work, what is the target function, what is the dataset, what denotes GA parents, how you select them, etc. you need to add more meaningful information in fig 11-12 accordingly, combining your own simulation.

5. It is very unsatisfactory in the results analysis part, the readability is poor. One can hardly read papers with incomplete axis titles, a simple legend with no meaning. Also, if you need to make a comparison with no other detailed explanation, just merge pictures like 18 and 19 to give a vivid presentation. Too simple analysis and introduction regarding the obtained figures in this section.

6. Steady-state optimization normally depend on parametric analysis and no further demand for AI algorithm, once it is regarded as a real-time dynamic optimization, It is also very important to show the response time when applying GA in the real application, for the algorithm response time to the shifting wind conditions is critical. Please add more introduction.

Minor:

1) Fig.1 please add axis titles. and BITA in contents.

2) title of section 4 is unclear.

3) suggested citing related journal papers from processes.

Author Response

(The authors gave the same response as above.)

Round 2

Reviewer 1 Report

The reviewer is satisfied with the revised manuscript and has no further questions for the authors. I recommend accepting this manuscript in its present form.

Author Response

Response to Reviewer #1:

Manuscript ID:  processes-2209104

Full Title:    Modeling of a Wind Power System Using the Genetic Algorithm Based on Doubly Fed Induction Generator for the Supply of Power to the Electrical Grid

Authors’ response to Reviewer #1:

Comments and Suggestions for Authors:

The reviewer is satisfied with the revised manuscript and has no further questions for the authors. I recommend accepting this manuscript in its present form.

Response:

I would like to thank the reviewer for all the corrections he made to us and they benefited us a lot in amending this article. I also thank him for accepting this manuscript for publication in your valuable journal.

Reviewer 3 Report

The paper has been revised accordingly, which presents more valuable information for readers. The topic is interesting and important for the large-scale use of wind power connecting to the power grid. 

One place that expects to be adjusted is in the conclusion part. don't use too arbitrary punctuation like ....,  please re-organize the last paragraph, into a more academic presenting structure.

Author Response

Response to Reviewer #3:

Manuscript ID:  processes-2209104

Full Title:    Modeling of a Wind Power System Using the Genetic Algorithm Based on Doubly Fed Induction Generator for the Supply of Power to the Electrical Grid

First of all, the authors wish to thank the editor and the reviewers for their efforts in reviewing the paper and for their valuable comments and recommendation given to improve the quality of the paper. The authors hope that the revised paper is improved to their satisfaction. Response to the specific comments and suggestions is given below. Revisions made in the manuscript are highlighted in the yellow color.

Authors’ response to Reviewer #3:

Comment #3 :

One place that expects to be adjusted is in the conclusion part. don't use too arbitrary punctuation like ....,  please re-organize the last paragraph, into a more academic presenting structure.

response:

    We thank the reviewer for these valuable comments and they have   been taken into consideration in amending this article.
